# FusionMaestro: Harmonizing Early Fusion, Late Fusion, and LLM Reasoning for Multi-Granular Table-Text Retrieval

## Abstract

Table-text retrieval aims to retrieve relevant tables and text to support open-domain question answering. Existing studies use either early or late fusion, but face limitations. Early fusion pre-aligns a table row with its associated passages, forming "stars," which often include irrelevant contexts and miss query-dependent relationships. Late fusion retrieves individual nodes, dynamically aligning them, but it risks missing relevant contexts. Both approaches also struggle with advanced reasoning tasks, such as column-wise aggregation and multi-hop reasoning. To address these issues, we propose `FusionMaestro`, which combines the strengths of both approaches. First, the *edge-based bipartite subgraph retrieval* identifies finer-grained edges between table segments and passages, effectively avoiding the inclusion of irrelevant contexts. Then, the *query-relevant node expansion* identifies the most promising nodes, dynamically retrieving relevant edges to grow the bipartite subgraph, minimizing the risk of missing important contexts. Lastly, the *star-based LLM refinement* performs logical inference at the star subgraph rather than the bipartite subgraph, supporting advanced reasoning tasks. Experimental results show that `FusionMaestro` outperforms state-of-the-art models with a significant improvement up to 42.6% and 39.9% in recall and nDCG, respectively, on the `OTT-QA` benchmark.

## 1 Introduction

Open-domain question answering (ODQA) over tables and text is important as it leverages the complementary strengths of structured and unstructured data. Tables offer vast amounts of related facts but lack diversity, while text provides broader contextual information (Chen et al., 2020b;a), making the integration of both modalities essential. Table-text retrieval, which retrieves question-relevant tables and text, is a key task in ODQA as it provides question-relevant context to readers of retriever-reader systems (Chen et al., 2020a; Huang et al., 2022; Ma et al., 2022; 2023; Kang et al., 2024).

Despite its importance, table-text retrieval faces two key challenges due to its multimodal nature. First, it involves resolving multi-hop relationships across diverse corpora for structured tables and textual passages (Chen et al., 2020a; Talmor et al., b). While textual data is generally unstructured and narrative-driven, tabular data is highly structured. Understanding its rows and columns involves interpreting structural semantics, making the integration of information from these two formats complex. Second, the retrieval process should support advanced reasoning for modality-specific operations such as column-wise aggregations and multi-modal operations like multi-hop reasoning.

Existing methods have achieved some success by employing either *early* or *late fusion* techniques in their top-$k$ retrieval. The *early fusion* strategy attempts to reduce the search space by grouping relevant documents before a query is presented. It pre-aligns a table row with associated passages via entity linking, creating a *fused block* as the retrieval unit (Chen et al., 2020a; Huang et al., 2022; Kang et al., 2024). In contrast, the *late fusion* strategy aligns relevant table rows and passages dynamically after the query is given. This alignment is typically driven by entity linking or query-based similarity matching. It returns a ranked sequence of evidence chains, where an *evidence chain* refers to a pair consisting of a table row and a passage (Ma et al., 2022; 2023).

However, the existing studies have several significant limitations.

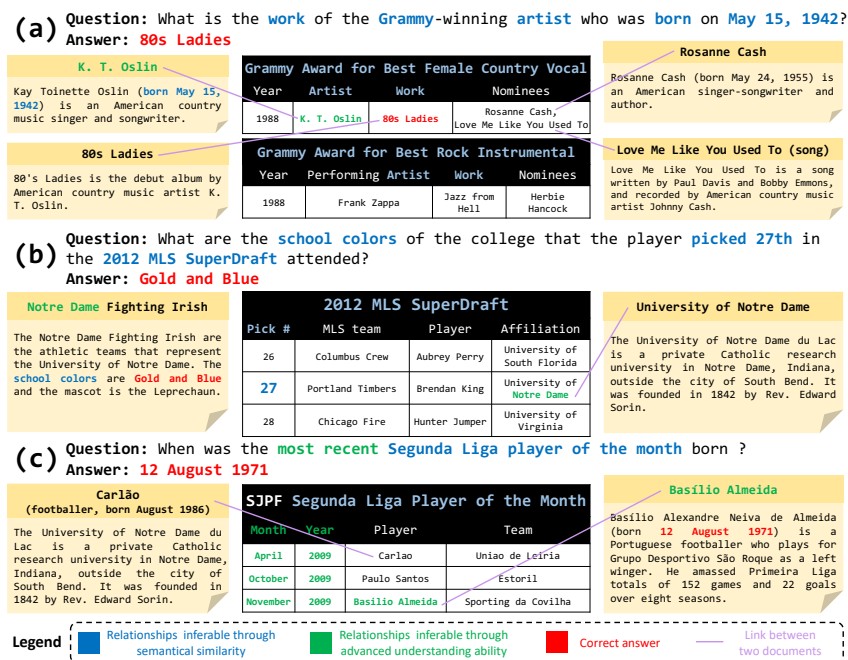

Figure 1: Simplified examples of three cases where existing methods struggle to retrieve the question-related documents correctly. (a) Inadequate granularity of retrieval units leading to inaccurate retrieval results. (b) Entity linking results cannot estimate essential query-aware relationships. (c) Inability of advanced reasoning such as table aggregation and multi-hop reasoning.

**(1) Inadequate granularity of retrieval unit.** The retrieval units used in the early fusion strategy are formed independently of the query, often including query-irrelevant passages. This problem leads to incorrect similarities between fused blocks and questions. For example in Figure 1(a), entity linking connects the `Grammy Award for Best Female Country Vocal` table to four surrounding passages. However, the irrelevant connections overwhelm the table, as the information related to `K. T. Oslin` is only essential information (Figure 1(a)). In the late fusion strategy, retrieving a single table segment or passage may be partially relevant to a query, incurring the risk of retrieving incorrect tables. For instance, during the first iteration of retrieval, the system might retrieve the `Grammy Award for Best Rock Instrumental` table instead of the correct one. Both this table and the correct one share partial relationships with the query, containing overlapping words such as `Grammy`, `Artist`, and `Work`, and can lead to confusion in identifying the correct target.

**(2) Missing query-dependent relationships.** The early fusion strategy relies on entity linking to predefine relationships between tables and passages. It fails to account for query-dependent links between documents that might contain the information necessary to answer the query. For instance, in Figure 1(b), the table `2012 MLS SuperDraft` is early fused with the entity `University of Notre Dame`. However, when the question specifies the information about `school colors`, it should be linked to the `Notre Dame Fighting Irish` passage.

**(3) Lack of advanced reasoning.** Queries that require complex reasoning, such as multi-hop or column-wise aggregation, often demand advanced logical inference beyond simple semantic similarity with the query. Since previous approaches rely on semantic similarity, they might fail to retrieve rows or passages identifiable through logical inference. For example, in Figure 1(c), the query involves understanding the `most recent Segunda Liga Player of the Month` is `Basilio Almeida`, where the row with the latest `Year` and `Month` combination has to be inferred.

We first formalize the terms proposed in previous studies using a *bipartite graph*, where table segments and passages are represented as two disjoint sets of nodes, and the links between them are represented as edges. Therefore, the term *fused block* used in the early fusion strategy (Chen et al., 2020a; Huang et al., 2022; Kang et al., 2024) can be represented as a star (Diestel, 2024) centered on a node of type `table segment`, with connected nodes of type `passage`. Similarly, the *evidence chain* used in the late fusion strategy (Ma et al., 2022; 2023) corresponds to an edge connecting a pair of nodes: one of type `table segment` and one of type `passage`.

In this paper, we propose `FusionMaestro`, a novel graph-based retrieval consisting of three stages: early fusion, late fusion, and LLM reasoning. Specifically, `FusionMaestro` adopts the following three key ideas:

**(1) Combined usage of early and late fusion.** We selectively leverage the advantages of both early fusion and late fusion. The early fusion stage provides comprehensive retrieval units by pre-aligning tables to their related passages before the query, mitigating the risk of retrieving incomplete or partially relevant information inherent in late fusion. Conversely, the late fusion stage dynamically captures query-dependent relationships during retrieval, addressing early fusion's reliance on predefined, query-independent links established via entity linking.

**(2) Graph refinement.** We leverage large language models (LLMs) to perform further advanced reasoning over the retrieved graph, enabling deeper logical inference beyond simple semantic similarity. For instance, in Figure 1(c), when the `SJPF Segunda Liga Player of the Month` table is retrieved, the LLM can perform aggregation to identify the most recent player and conduct multi-hop reasoning to select the corresponding passage for `Basilio Almeida`.

**(3) Granularity determination for each retrieval stage.** In our retrieval pipeline, each stage - early fusion, late fusion, and graph refinement - serves a distinct purpose, necessitating the precise determination of the appropriate operational units for each. For the early fusion stage, we propose a novel edge-level retrieval mechanism, which balances the challenge of excluding query-irrelevant context in star graph retrieval and avoiding the partial information problem in node-based retrieval. In the late fusion stage, we set the unit as an individual node. We identify query-relevant nodes within the graph produced by the early fusion stage so that we can design the late fusion process to expand the graph using only nodes closely aligned with the query context. This approach mitigates the challenge where the earlier stage may retrieve nodes irrelevant to the query. Finally, the graph refinement stage provides the fully expanded graph from late fusion to the LLM, which can increase hallucination risks due to the inclusion of unnecessary nodes. To mitigate this, we decompose the graph into smaller star graphs.

Experimental results demonstrate that `FusionMaestro` significantly outperforms state-of-the-art systems, with a 42.6% improvement in AR@2 and a 39.9% improvement in nDCG@50.

## 2 Related Work

### 2.1 Open-domain Question Answering

Open-Domain Question Answering (ODQA) is the task aimed at answering factual questions using a large knowledge corpus (Zhang et al., 2023). Representative ODQA benchmarks such as Natural Questions (Kwiatkowski et al., 2019), TriviaQA (Joshi et al., 2017), and SearchQA (Dunn et al., 2017) consist of single-hop queries that require information found in a single passage within a corpus of unstructured texts. Further advances were shown by HotpotQA (Yang et al., 2018) and WikiHop (Welbl et al., 2018), presenting challenging queries that require multi-hop reasoning across multiple passages. However, these benchmarks support only unstructured passages and do not consider multi-hop reasoning across structured tables and unstructured passages, which is essential in table-text retrieval tasks. `OTT-QA` (Chen et al., 2020a) is the first ODQA benchmark that supports multi-hop reasoning between tables and text. It introduces questions that require reasoning over both tables and their associated passages, providing a more realistic and challenging scenario for retrieval methods.

### 2.2 Table-Text Retrieval

Table-text retrieval methods can be broadly categorized into early fusion and late fusion approaches. These terms, initially used in multimodal tasks like image-sentence retrieval and semantic video analysis, describe whether different modalities are encoded jointly or separately (Wang et al., 2022; Snoek et al., 2005; Gadzicki et al., 2020). Similarly in the context of table-text retrieval, early and late fusion approaches differ based on whether tables and text are linked before or after the retrieval process (Kang et al., 2024).

*Early fusion* approaches (Chen et al., 2020a; Huang et al., 2022; Kang et al., 2024), before the query is given, connect each table row with its related passages using entity linking, forming 'fused blocks' that serve as basic retrieval units. These fused blocks are later retrieved during online time by measuring their similarity to the query. Early fusion approaches show two limitations: (i) The fused blocks may include numerous query-irrelevant passages since they include all passages linked to a

row, without considering the query. This large retrieval unit not only introduces unrelated passages into the retrieval results but also increases information loss when encoding the embeddings for the fused blocks, thereby reducing the overall retrieval accuracy. (ii) Offline-generated fused blocks are unable to consider query-dependent relationships that must be resolved online, as illustrated in Figure 1(b). Our retrieval method addresses limitation (i) by using *edges* as basic units of retrieval, as they are more fine-grained than fused blocks. Furthermore, to address limitation (ii), we propose a query-relevant node expansion that adds query-dependent relationships online.

Late fusion approaches (Ma et al., 2022; 2023) dynamically group relevant documents online. They begin with retrieving table segments relevant to the question, followed by retrieving passages associated with these segments to establish connections between the documents. These methods require considering all possible pairs between table segments and passages online, resulting in a vast search space. Search algorithms like beam search are employed to address this problem, but can lead to an error propagation problem as retrieving a single table segment or passage may contain only partial relevant information. Our approach utilizes an edge-based retrieval, which captures richer context by connecting table segments and passages, enabling a more accurate seed document retrieval.

*Both* early fusion and late fusion approaches predominantly rely on semantic similarity for retrieval. Therefore, it may fail to retrieve table segments and passages that require advanced reasoning (e.g., column-wise aggregation, multi-hop reasoning) to be found, as shown in Figure 1(c). To address the limitations, we propose a star-based LLM refinement, which leverages the logical inference ability of LLM to refine the retrieved results using advanced reasoning.

Additionally, our retrieval method applies different levels of granularity (e.g., edges, nodes, stars) tailored to each retrieval phase. DRAMA (Yuan et al., 2024) also adopts a multi-granularity approach but is limited to a constrained setting where relevant tables and passages are provided, unlike our method, which operates in an open-domain context. GTR (Wang et al., 2021) and MGNETS (Chen et al., 2021) focus on enhancing table encoding using graph-based methods, whereas our work targets bridging semantic relationships between tables and text in open-domain retrieval.

# 3 PRELIMINARIES

## 3.1 PROBLEM FORMULATION

Table-text retrieval is involved from a retrieval *corpus* $\mathcal{C}$, which comprises two distinct sets: a collection of passages $\mathcal{C}_P = \{P^{(1)}, \ldots, P^{(n)}\}$ and a collection of tables $\mathcal{C}_T = \{T^{(1)}, \ldots, T^{(m)}\}$. A *passage* is defined as a sequence of tokens $P$, representing unstructured text. A *table* is a structured matrix $T$, consisting of cells $T_{i,j}$, where $i$ and $j$ indicate the row index and the column index, respectively. Each cell $T_{i,j}$ may contain a number, date, phrase, or sentence. We define a document as either a passage or a table. Given a query $q$, the objective of table-text retrieval is to retrieve from corpus $\mathcal{C}$ a ranked list of documents such that the document containing the answer span $a$ is positioned among the top results.

We split a table into multiple table segments, as commonly used in existing studies. Because a single table can easily exceed the token limits of language models, a table $T$ is combined with its header to form a list of table segments $T = [S^{(1)}, \ldots, S^{(m')}]$ (Chen et al. 2020a). This process results in (i) a corpus $\mathcal{C}$ composed of table segments $\mathcal{C}_S$ and passages $\mathcal{C}_P$ (i.e., $\mathcal{C} = \mathcal{C}_S \cup \mathcal{C}_P$) and (ii) a mapping $\mathcal{M} : \mathcal{C}_S \to \mathcal{C}_T$ to associate table segments with their original table.

## 3.2 TABLE-TEXT RETRIEVAL AS BIPARTITE GRAPH RETRIEVAL

We adopt a graph representation, denoted by $G = (V, E, \Phi, \Gamma, \Lambda)$, to generalize various methods used in existing studies. Here, $V$ is the set of vertices corresponding to a table segment or a passage, and $E$ is the set of edges representing relationships between (table segment, passage) pairs. The mapping $\Phi : V \to \{\texttt{table segment}, \texttt{passage}\}$ maps each node to its type, while $\Gamma$ maps a node to its attributes, such as the text of a passage or the matrix of table structures. The mapping $\Lambda : E \to \mathbb{R}$ maps each edge to its score.

The corpus can be expressed as the initial graph $G_{init} = (V_{init}, \emptyset, \Phi, \Gamma, \Lambda_{init})$, where each node in $V_{init}$ one-to-one corresponds to a table segment or a passage in $\mathcal{C}$. Early fusion generates table-text relationships via entity linking and updates $G_{init}$ before a query $q$ is presented. Given a query $q$, late fusion dynamically generates query-dependent table-text relationships to update $G_{init}$. Finally, we

aim to retrieve a query-relevant edge-scored bipartite graph $G_q = (V_q, E_q, \Phi, \Gamma, \Lambda_q)$ from $G_{init}$. This problem is often interpreted as finding a ranked sequence of edges $\mathcal{E}_q$ from all possible edges, as the retrieved results are fed to a reader with limited context size (Ma et al., 2022; 2023). $\mathcal{E}_q$ is often generated by sorting each edge $e$ in $G$ using its edge scores $\Lambda(e)$.

# 4 PROPOSED METHOD

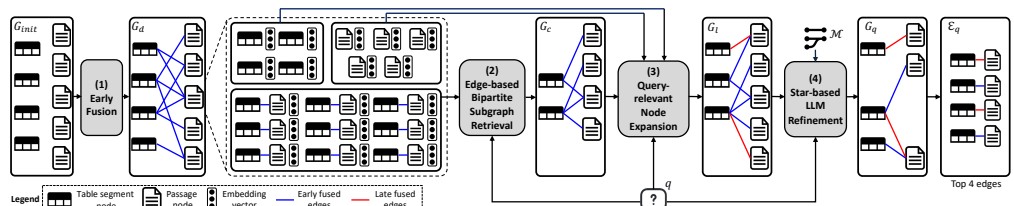

Figure 2: Overview of `FusionMaestro`: (1) The initial graph $G_{init}$ is first early fused to generate a graph $G_d$. Each node and edge of $G_d$ are embedded. (2) The edges of $G_d$ are retrieved using the query $q$, then integrated into a candidate bipartite subgraph $G_c$. (3) The most query-relevant nodes in $G_c$ are identified as seed nodes. Nodes from $G_{init}$ that are relevant to both the seed node and the query are expanded into $G_c$, forming the expanded graph $G_l$. (4) LLM performs aggregation over restored tables to identify new relevant table rows, and then eliminates irrelevant passages.

**Overview.** We propose `FusionMaestro`, a novel graph-based retrieval to leverage the advantages of both early and late fusion. It operates in three main stages as follows.

**Edge-based Bipartite Subgraph Retrieval.** It uses edges as the basic retrieval unit on a bipartite data graph generated by early fusion. It searches for edges relevant to the query within the bipartite graph and integrates the retrieved edges into a single bipartite subgraph. The retrieval unit is set to an edge in this process. This enables a more accurate retrieval of query-relevant subgraphs as it is less likely to contain query-irrelevant nodes like fusion blocks, and it also provides richer context than a single document.

**Query-relevant Node Expansion.** It reinforces the retrieved bipartite subgraph with new nodes found by performing an additional hop (i.e., expansion) from the input subgraph. In this step, we first identify the most promising nodes within the subgraph to perform an additional hop from. These nodes are called seed nodes. Next, the seed nodes are combined with the query to generate expanded queries, which are then used to find the candidate nodes to be expanded to the graph. Lastly, the most promising ones among the candidates are actually expanded into the subgraph.

**Star-based LLM Refinement.** It further refines the expanded graph via LLM's advanced reasoning, such as aggregation or multi-hop reasoning. It first restores the original tables of the table segments within the expanded graph, then performs an aggregation operation if a query contains one. The output table segments are then added to the graph, along with its related passages. Second, it verifies whether each edge in the graph is relevant to the query in a star-graph-wise manner. The edges verified as irrelevant are excluded from the refined graph. Lastly, it decomposes the refined graph into a ranked sequence of edges.

## 4.1 EDGE-BASED BIPARTITE SUBGRAPH RETRIEVAL

`FusionMaestro` initiates its process with the retrieval of a bipartite subgraph through two key steps: early fusion and edge retrieval. (i) In the early fusion step, a bipartite data graph $G_d$ is generated from $G_{init}$ by linking table segments and passages via entity linking. Each edge in this graph represents a meaningful connection between the two passages of different modalities. An embedding is also computed for each edge in this step. (ii) In the edge retrieval step, the set of most query-relevant edges is identified by leveraging the semantic similarity between the query and the edge embeddings. It is then integrated to construct the candidate bipartite subgraph $G_c \subset G_d$.

The early fusion step starts by generating edges from the initial graph $G_{init}$ that has no edges. We follow the prior methods for the edge generation process, a two-step process of entity linking followed by entity recognition (Ma et al., 2022; 2023). The output graph of this process $G_d$ is a bipartite graph, since the edges are generated only between passage nodes and table segment nodes.

The next step is to generate embeddings for each edge. Previous early fusion techniques tend to create an embedding for each star graph, and the embeddings share a fixed number of vectors. Here, a star graph is a graph where one table segment node is linked to multiple connected passages. We introduce a fine-grained approach that generates token-level embeddings at the edge level, aiming to balance providing richer information and minimizing information loss. We also adopt a late interaction model (Santhanam et al., 2021) that dynamically adjusts the length of the embedded vector sequence, preserving fine-grained token-level details.

The generated edges are then embedded into a sequence of vectors. They first are linearized into a token sequence as follows.

$$x = [x_1, ..., x_{l_x}] = [\, Linearize(\Gamma(S)); \; \Gamma(P) \,] \quad e = (S, P) \tag{1}$$

where $x$ is the resulting token sequence representing the edge, and $l_x$ denotes the length of this sequence. $x$ is then embedded into a sequence of vectors. Mathematically, the encoding of both the query $q$ and the token sequence $x$ can be expressed as:

$$\mathbf{Q} = f_e(q) \in R^{l_q \times d}; \quad \mathbf{X} = f_e(x) \in \mathbb{R}^{l_x \times d} \tag{2}$$

based on the principles of `ColBERTv2` (Santhanam et al. 2021). $l_q$ represents the length of the query and $f_e$ is our late interaction edge encoder.

At online time, a bipartite candidate subgraph is generated by retrieving and merging the offline-embedded edges. This step is conducted through a three-stage process. First, we apply an initial retrieval where the late interaction edge encoder $f_e$ computes the similarity scores between the query $q$ and each edge $e$. The similarity is calculated as:

$$f_e(q, x) = \sum_{i=1}^{l_q} \max_{j \in [1, l_x]} \mathbf{Q}_i \mathbf{X}_j \tag{3}$$

This score quantifies the degree of alignment between the query tokens and the tokens in the edge. The top-$k_1$ edges are selected based on these scores. In the second stage, these query-edge pairs are passed through an all-to-all interaction reranker $g_e$, which performs a more detailed similarity evaluation. This identifies the most contextually relevant edges, allowing us to identify the top-$k_2$ query-relevant edges ($k_2 < k_1$). Finally, the $k_2$ edges are integrated into the bipartite subgraph $G_c = (V_c, E_c, \Phi, \Gamma, \Lambda_c)$, forming the candidate bipartite subgraph that serves as the foundation for further expansion and refinement. The score for each generated edge $e$ is saved as $g_e(e)$ in the score mapping $\Lambda_c$. $\Lambda_c$ will not be used until Section 4.3 where the final ranked list of edges will be generated.

The late interaction encoder $f_e$ was fine-tuned following the training scheme of `ColBERTv2` (Santhanam et al., 2021), and the all-to-all interaction reranker $g_e$ was fine-tuned using contrastive loss. Detailed explanations of the fine-tuning process, including the construction of the training dataset, can be found in Appendix § B.1.

## 4.2 QUERY-RELEVANT NODE EXPANSION

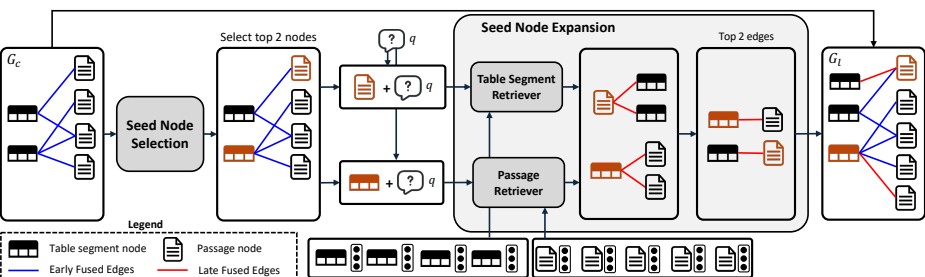

Figure 3: The overall procedure of query-relevant node expansion. The beam width $b$ is set as 2 in this example. The orange-colored nodes indicate the selected seed nodes.

The query-relevant node expansion process identifies query-relevant edges at the node level based on late fusion. This on-the-fly expansion generates graph $G_l$, a graph that includes additional expanded nodes, from $G_c$. We perform the expansion process at the node level, which is the most fine-grained

level. This is to address the issue that early fusion inevitably includes query-irrelevant nodes in the candidate subgraph as the fused blocks are determined independent of the query. Formally, the node-based expansion process can be expressed as finding a set of edges that meet the following objective function.

$$\underset{(u,v)\in E^*\wedge u\in V_c}{\arg\max}\ p(u,v|q) = p(v|u,q)p(u|q) \tag{4}$$

Here, $u$ represents a node in the candidate graph $G_c$, and $v$ is a node adjacent to $u$ in the complete bipartite graph $G^*$. The complete bipartite graph $G^* = (\mathcal{C}_{init}, E^*, \Phi, \Gamma, \Lambda_{init})$ contains all possible edges between table segments and passages.

We employ a two-step beam search to identify expanded edges.

**(1) Seed node selection**: From the candidate bipartite subgraph $G_c$, we select a set of nodes that contain information most relevant to the query. This corresponds to finding the set of nodes that show the highest $p(u|q)$.

**(2) Seed node expansion**: For each seed node, we iterate through its neighbors in the complete bipartite graph $G^*$, calculating the similarities between each expanding node and the pair of the query and the seed node. Among these, node pairs that exhibit the highest similarity with the query are returned as edges, further expanding the graph $G_c$ to $G_l$ by adding these edges.

The seed node selection calculates $p(u|q)$ for each $u \in V_c$ to identify the top-$b$ (i.e., beam width) nodes that contain the most relevant information to the query. The probability $p(u|q)$ is determined by calculating the semantic similarity between the query and each node $u$ in $G_c$, which is normalized using a softmax function. This similarity is computed through an all-to-all interaction-based node reranker $g_n$. The similarity scores $g_n([q; \Gamma(u)])$ for all $u$ are used to select the top-$b$ seed nodes.

The seed node expansion computes $p(v|u,q)$ for each node $v$ connected to seed node $u$ in the complete bipartite graph $G^*$. These conditional probabilities are calculated using the expanded query retrieval technique (Xiong et al.). In this technique, the score function is expressed with the expanded query as $s([q; \Gamma(u)], v)$, and it is calculated by two late interaction models: $f_{P\to S}$ for a table-segment-typed expanding node and a passage-typed seed node, and $f_{S\to P}$ for the opposite. The calculated scores are normalized using a softmax function to compute $p(v|u,q)$. We then calculate $p(u,v|q)$ using Equation 4. The final probability is used to select the top-$b$ probable edges from the pairs of seed node $u$ and expanding node $v$. They are added to $V_c$ and $E_c$ of $G_c$, forming the updated bipartite graph $G_l = (V_l, E_l, \Phi, \Gamma, \Lambda_l)$. Each new edge is scored using the identical scoring module $g_e$ discussed earlier in § 4.1.

The change in retrieval accuracy of `FusionMaestro` based on beam width $b$ is discussed in Appendix § C.2. The node reranker $g_n$ are fine-tuned for node selection. The late interaction encoders $f_{P\to S}$ and $f_{S\to P}$ used for expanded query retrieval are fine-tuned following the `ColBERTv2` training scheme (Santhanam et al., 2021). Detailed explanations for fine-tuning, including the construction of the training dataset, for all the modules can be found in Appendix § B.2 and Appendix § B.3.

### 4.3 STAR-BASED LLM REFINEMENT

Traditional semantic similarity is insufficient to correctly retrieve documents for queries that require logical inference, such as column-wise aggregation or multi-hop reasoning. To overcome this problem, we leverage the advanced reasoning capabilities of large language models (LLMs), which allow us to refine the retrieved result through logical inference. The main goal is to use LLM's logical inference to add relevant edges to the graph $G_l$ and remove irrelevant ones.

It is non-trivial to choose the specific format or unit of providing the graph $G_l$ to the LLM. We considered two approaches: including the entire graph $G_l$ in a single prompt to return the relevant set of nodes, and decomposing $G_l$ into star graphs, with each star graph generating its own set of relevant nodes. Among these, using star graphs as the unit proved to be 12.4% more effective, leading us to select this as our unit for logical inference (§ 5.4).

The refinement process occurs in two phases: *column-wise aggregation* and *passage verification*. The column-wise aggregation step restores tables from table segments, then identifies the candidate rows based on the query and adds them back to the graph. The passage verification step evaluates each star graph, returning the passages essential for answering the query. The refined edge-scored graph is then decomposed into a ranked sequence of edges to produce the final retrieval output. A detailed process is expressed in Figure 4.

**Column-wise Aggregation.** It aims to accurately infer the correct result rows for table aggregation operations, as exemplified in Figure 1(c). In our early fused graph $G_d$, tables are divided into individual row-based table segments, making it hard for the previous-stage retrievers to perform proper aggregation. It becomes necessary to reconstruct the original tables and perform reasoning over these full tables.

Since not every query requires aggregation, the first step is to prompt the LLM to determine whether the input query necessitates an aggregation operation. If the query is classified as an aggregation query, the process follows two steps: (i) Table restoration: For each table segment, we utilize the mapping function $\mathcal{M}$ to restore the original table. (ii) Aggregation: The restored tables are provided to the LLM in the format of star graph, where LLM performs the aggregation and returns the rows corresponding to the aggregation result. The returned rows are subsequently added back to $G_l$ along with their associated passages to generate $G_a$.

**Passage Verification.** It aims to leverage the LLM's logical inference capabilities to remove the query-irrelevant passages within $G_a$. Similar to the column-wise aggregation step, we provide $G_a$ to the LLM in the form of star graphs, units that contain multi-hop relationships while not exceeding the context limit. The LLM performs a binary verification to determine whether each edge is relevant to the query, without recalculating their scores. As a result, query-irrelevant edges are eliminated, leaving a refined, edge-scored graph $G_q$. Examples of prompts used for this step can be found in Appendix § E.

**Top-K Edge Selection.** The graph $G_q$ is then transformed into a ranked sequence of edges $\mathcal{E}_q$ by applying the score mapping $\Lambda_c$. Specifically, all edges $e$ in $G_q$ are ranked in descending order based on their scores $\Lambda_c(e)$.

## 5 EXPERIMENTS

### 5.1 EXPERIMENT SETUP

**Hardware and Software Settings.** We conducted our experiments on a machine with Intel(R) Xeon(R) Gold 6230 CPU @ 2.10GHz CPU and 1.5T of RAM with the OS of Ubuntu 22.04.4 and 4 RTX A6000 GPUs.

**Compared Techniques.** `FusionMaestro` is compared with the SOTA methods. The early fusion methods include Fusion-Retriever (Chen et al., 2020a), OTTeR (Huang et al., 2022), and DoTTeR (Kang et al., 2024). While, the late fusion approaches include Iterative-Retriever (Chen et al., 2020a), CORE (Ma et al., 2022), and COS (Ma et al., 2023).

**Datasets.** We conducted our experiments using two datasets: `OTT-QA` (Chen et al., 2020a) and MultimodalQA (`MMQA`) (Talmor et al., a). `OTT-QA` serves as the primary dataset for comparison, as it is the only dataset specifically designed for open-domain QA involving both tables and texts. The `OTT-QA` corpus contains 400K tables and 5M passages, and it is composed of a training set with 42K question-answer pairs, along with development and test sets of 2K question-answer pairs each. `MMQA` is a QA dataset for multi-hop reasoning over images, passages, and tables. Though it does not align perfectly with our task's requirements, it was utilized as a supplementary dataset to test the generalizability of our method. The `MMQA` corpus includes 10K tables and 210K passages, with a development set of 1.3K question-answer pairs. We excluded image-based questions and conducted experiments in an open-domain setting using the entire corpus, without utilizing the reference candidates provided for each question.

### 5.2 MAIN RESULTS

We evaluated the accuracy of the retrieved documents using top-$k$ Answer Recall (AR@$k$), nDCG@$k$, and Hits@4K as well as the end-to-end performance measured by EM and F1 scores. AR@$k$ measures the percentage of queries where the correct answer string appears within the top-$k$ retrieved edges (Ma et al., 2023). nDCG@$k$ measures the ranking quality of the retrieved edges up to position $k$, depending on each edge's relevance to the query and its position in the ranked list. Hits@4K measures the proportion of cases where the answer span exists within the top 4096 tokens after linearizing the sequence of ranked edges (Chen et al., 2020a). Additionally, we perform end-to-end question-answering experiments to evaluate how retrieval accuracy impacts overall QA performance, using exact match (EM) accuracy and F1 score to assess the quality of generated answer spans. If the number of edges in $\mathcal{E}_q$ is fewer than the target edges, we include the edges

Table 1: Retrieval accuracy on `OTT-QA`'s dev set for `FusionMaestro` and six competitors. Results marked with † indicate reproduced values.

| Model | AR@2 | AR@5 | AR@10 | AR@20 | AR@50 | nDCG@50 | HITS@4k |
|---|---|---|---|---|---|---|---|
| Iterative Retriever | – | – | – | – | – | – | 27.2 |
| Fusion Retriever | – | – | – | – | – | – | 52.4 |
| OTTeR† | 31.4 | 49.7 | 62.0 | 71.8 | 82.0 | 25.9 | 70.1 |
| DoTTeR† | 31.5 | 51.0 | 61.5 | 71.9 | 80.8 | 26.7 | 70.3 |
| CORE† | 35.3 | 50.7 | 63.1 | 74.5 | 83.1 | 25.4 | 77.2 |
| COS† | 44.4 | 61.6 | 70.8 | 79.5 | 87.8 | 33.6 | 81.8 |
| FusionMaestro | **63.3** | **76.7** | **85.0** | **90.4** | **94.2** | **47.0** | **91.8** |

Table 2: AR@$k$ on `MMQA`'s dev set for `FusionMaestro` and `COS`. Results marked with † indicate reproduced values.

| Model | AR@2 | AR@5 | AR@10 | AR@20 | AR@50 |
|---|---|---|---|---|---|
| COS† | 50.7 | 59.7 | 67.1 | 72.4 | 79.5 |
| FusionMaestro | **70.5** | **77.8** | **81.0** | **82.6** | **86.2** |

removed during the star-based LLM refinement stage to assess the retrieval accuracy. The values of $k \in \{2, 5, 10, 20, 50\}$ were selected based on the $k$-values used in the evaluation of SOTA early and late fusion models (Kang et al., 2024; Ma et al., 2023).

We evaluated the retrieval accuracy of `FusionMaestro` on the OTT-QA and MMQA datasets. For the OTT-QA dev set, we measured AR@$k$, nDCG@$k$, and Hits@4K across `FusionMaestro` and six competitors, and the results summarized in Table 1. `FusionMaestro` consistently outperforms other retrievers across different $k$ values ($k \in 2, 5, 10, 20, 50$). It outperforms the state-of-the-art `COS` model by an average of 19.0% in AR, with the performance gap widening as $k$ decreases. At $k = 2$, `FusionMaestro` achieves as much as 42.6% higher answer recall than `COS`. This improvement is further reflected in nDCG@50, where `FusionMaestro` exhibits a 39.9% gain. Additionally, the Hits@4K metric shows a 12.2% improvement over `COS`. To assess the generalizability of `FusionMaestro`, we extended our evaluation to the `MMQA` dataset, comparing its AR@$k$ performance against `COS`. As detailed in Table 2, `FusionMaestro` maintains its superior performance across all $k$ values, achieving an average improvement of 20.9% in AR across all $k$ values, further reinforcing its robustness across different datasets.

Table 3: End-to-end question answering accuracy for development and test set of `OTT-QA`.

| Algorithm | Dev | | Test | |
|---|---|---|---|---|
| | EM | F1 | EM | F1 |
| OTTeR | 37.1 | 42.8 | 37.3 | 43.1 |
| DoTTeR | 37.8 | 43.9 | 35.9 | 42.0 |
| CORE | 49.0 | 55.7 | 47.3 | 54.1 |
| COS | 56.9 | 63.2 | 54.9 | 61.5 |
| FusionMaestro | **59.3** | **65.8** | **57.0** | **64.3** |

To assess the impact of our retrieved results on the reading task, we evaluated the end-to-end question-answering performance of `FusionMaestro` and `COS` on `OTT-QA`'s dev and test sets. The results are shown in Table 3. As for the reader, we followed `COS` to employ a Fusion-in-Encoder (FiE) model (Kedia et al., 2022) fine-tuned on the `OTT-QA` dataset. To ensure a fair comparison, we provided 50 edges as input to the reader, following the evaluation protocol used by `COS`. The results indicate that compared to the `COS` model, our approach improved both EM and F1 scores by 4.2% and 4.1% on the development set, as well as by 3.8% and 4.6% on the test set, respectively. This demonstrates that the well-retrieved documents from our algorithm effectively assist the reader in generating more accurate answers.

## 5.3 ABLATION STUDY

We performed an ablation study to assess the contribution of query-relevant node expansion (QNE) and star-based LLM refinement (SLR) to retrieval accuracy. We implemented two additional versions of `FusionMaestro`. In one version of `w/o QNE`, we removed the QNE module and `FusionMaestro` passes the candidate bipartite subgraph $G_c$ directly to the SLR module. In the

Table 4: Retrieval accuracy of `OTT-QA`'s development set for `FusionMaestro`'s various design factors (`QNE` = Query-relevant Node Expansion, `SLR` = Star-based LLM Refinement).

| Algorithm | AR@2 | AR@5 | AR@10 | AR@20 | AR@50 | nDCG@50 | EM | F1 |
|---|---|---|---|---|---|---|---|---|
| FusionMaestro | **63.3** | **76.7** | **85.0** | **90.4** | 94.2 | **47.0** | 59.3 | 65.8 |
| w/o QNE | 62.5 | 74.7 | 82.7 | 88.4 | 92.7 | 45.1 | 56.9 | 63.2 |
| w/o SLR | 60.0 | 75.2 | 84.7 | 90.1 | **94.6** | 46.5 | 59.0 | 65.7 |

other of `w/o SLR`, the SLR module was removed and `FusionMaestro` decomposes the expanded graph $G_l$ into a list of edges.

As in Table 4, we found that removing the QNE module led to an average performance degradation of 2.1% in AR across all $k$ values and 4.2% in nDCG@50. This highlights the role of QNE in generating query-relevant edges missed by offline entity linking. Secondly, for the `w/o SLR` algorithm, we observed a noticeable drop in AR@2, AR@5, AR@10, AR@20, and nDCG@50, with accuracy decreases of 5.5%, 2.0%, 0.4%, 0.3%, and 1.1%, respectively. This suggests that LLM-based node selection helps accurately identify the query-relevant nodes in complex queries where logical inference is needed. This tendency is particularly evident when $k$ is small. Interestingly, for AR@50, the `w/o SLR` version slightly outperformed `FusionMaestro` by 0.4%. This phenomenon can be attributed to LLM hallucinations. In some cases, `FusionMaestro`'s SLR module failed to select the correct query-relevant nodes. We present the qualitative analysis results in Appendix § D.

### 5.4 IMPACT OF GRANULARITY TO ACCURACY

We investigated the impact of retrieval unit granularity on accuracy by comparing three versions of our subgraph retriever module, each with a distinct type of retrieval unit. (i) Node: it retrieves table segments first then links the related passages via entity linking. (ii) Star graph: it retrieves the star graphs and then integrates them into a graph. (iii) Edge: it retrieves the edges, and integrates them to generate a graph. For a fair comparison, we conducted experiments using the `ColBERTv2` baseline model without fine-tuning it for each retrieval unit.

Table 5: Comparison between star-graph-based search and edge-based search.

| Retrieval Unit | AR@2 | AR@5 | AR@10 | AR@20 | AR@50 | nDCG@50 |
|---|---|---|---|---|---|---|
| Node | 29.3 | 47.4 | 58.8 | 68.5 | 79.5 | 23.8 |
| Star Graph | 37.9 | 57.4 | 66.9 | 76.4 | 84.5 | 28.5 |
| Edge | 49.1 | 63.1 | 70.6 | 77.6 | 85.1 | 34.2 |

As shown in Table 5, the edge-based retrieval consistently outperformed the others. On average across all values of $k$, edge-based retrieval outperformed star graph-based and node-based retrieval by 6.9% and 12.4%, respectively. In terms of nDCG@50, it outperformed them by 20% and 43.7%, respectively. This highlights edge-based retrieval's ability to provide richer information while minimizing information loss, striking an effective balance compared to the other methods. Additionally, we conducted experiments using two refinement units with an LLM, comparing the performance of a full graph prompt versus individual prompts for each star graph. The star graph setting, which reduced irrelevant information in prompts, achieved an nDCG@50 score 12.4% higher than the full graph setting (41.8), demonstrating improved performance and a reduced risk of hallucinations.

## 6 CONCLUSION

We presented `FusionMaestro`, a novel method for table-text retrieval that harmonizes the strengths of both early fusion and late fusion techniques while incorporating large language model (LLM) reasoning. It addresses the limitations of existing approaches by introducing a multi-granular retrieval pipeline that optimally balances granularity across retrieval stages. By employing edge-based bipartite subgraph retrieval, query-relevant node expansion, and star-based LLM refinement, `FusionMaestro` provides more accurate retrieval by dynamically constructing query-relevant bipartite graphs. Experimental results demonstrate that `FusionMaestro` significantly outperforms state-of-the-art models, with a 42.6% improvement in AR@2 and a 39.9% gain in nDCG@50, on the OTT-QA benchmark.

**Reproducibility Statement**  We provide prompt examples for operations performed in star-based LLM refinement, including aggregation query classification in Appendix E.1, column-wise aggregation in Appendix E.2, and passage verification in Appendix E.3. Additionally, OTTeR and DoTTeR were reproduced using the official code available at `OTTeR` and `DoTTeR`, respectively. COS and CORE were reproduced using the official code from `UDT-QA`. The source code, data, and other artifacts for `FusionMaestro` have been made available at `anonymous.4open.science`.

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

## A   STAR-BASED LLM REFINEMENT SUPPLEMENTARY

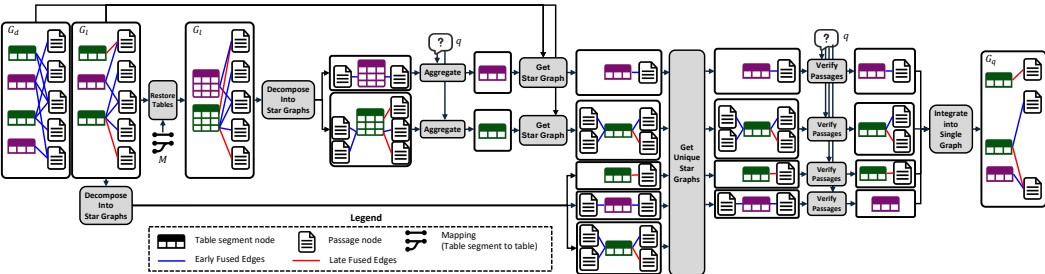

Figure 4: The overall process of star-based LLM refinement for queries classified as aggregation queries. Table segment nodes of the same color (green, purple) indicate segments that belong to the same original table.

## B   TRAINING SCHEME

### B.1   EDGE RETRIEVER AND RERANKER

The training scheme for our encoder $f_e$ follows the methodology outlined in `ColBERTv2` (Santhanam et al. 2021), leveraging a combination of in-batch negative loss and knowledge distillation loss to train the model. Specifically, the in-batch negative loss treats the edges corresponding to other queries within the same batch as negative samples. This approach calculates a contrastive loss between the positive and negative edges. In constructing the training dataset, it is crucial to have both positive and negative edges for each query. To define the positive edge, we use passages containing the answer and the associated table segments as ground truth and denoted as $x_{gt}$. Conversely, negative edges are constructed by combining hard negative tables and passages from prior work (Ma et al. 2023) with in-batch negative edges and are denoted as $n(q)$. The contrastive loss $L_{cl}$ is represented as follows:

$$L_{cl} = - \sum_{(q,x_{gt})} \log \frac{exp(s(q, x_{gt}))}{exp(s(q, x_{gt})) + \sum_{z \in n(q)} exp(s(q, z))} \quad (5)$$

The knowledge distillation process refines the edge encoder using a teacher-student model setup. The distillation loss is computed based on the KL divergence between the score distribution generated by the teacher model and the training encoder.

Here, the teacher model is the all-to-all interaction reranker $g_e$ fine-tuned with the contrastive loss, which serves as a more precise reference for edge relevance. This method ensures that the encoder learns from a more sophisticated model, improving its capacity to accurately rank edges based on the query.

### B.2   NODE RERANKER

The training method for the node reranker $g_n$ is identical to that of the edge reranker $g_e$. For constructing the training dataset, we utilize the `OTT-QA` dataset (Chen et al., 2020a). Positive nodes are defined as those directly connected to the nodes that contain the correct answer in `OTT-QA`. In contrast, negative nodes are selected from the set of nodes retrieved through edge-based bipartite subgraph retrieval, excluding any nodes connected to the answer-containing nodes.

### B.3   EXPANDED QUERY RETRIEVERS

The training scheme for our expanded query retrievers $f_{S \to P}$, $f_{P \to S}$ also follows the methodology outlined in `ColBERTv2` (Santhanam et al. 2021). To construct the training dataset, we generated triples consisting of the expanded query, positive node, and negative node. Expanded queries were created by incorporating nodes that are connected to the node containing the answer. Positive nodes

consist of the nodes that contain the answer. Negative nodes are constructed using hard negative nodes as outlined in prior work (Ma et al. 2023).

## C EXPERIMENT SUPPLEMENTARIES

### C.1 IMPLEMENTATION DETAILS

In our edge generation step (§ 4.1), we used the same named entity recognition and entity linking models used by COS (Ma et al., 2023). For the late-interaction edge retriever $f_e$ (§ 4.1) and the expanded query retrievers $f_{P \to S}$ and $f_{S \to P}$ (§ 4.2), we employed ColBERTv2 (Santhanam et al., 2021) as the baseline model. For the all-to-all interaction edge reranker $g_e$ (§ 4.1) and node reranker $g_n$ (§ 4.2), we used the bge-reranker-v2-minicpm-layerwise (BAAI, 2024), specifically utilizing layer 24 as the baseline model. Lastly, for star-based LLM refinement (§ 4.3), we used Llama-3.1-8B-Instruct (Dubey et al., 2024) as the large language model. In our experiments, The value of $k_1$ for the edge retriever $f_e$ was set to 400. Since COS selects the top-200 nodes as seed nodes, we fixed $k_2$ for the edge reranker $g_e$ to 100 to ensure a fair comparison.

### C.2 PARAMETER SENSITIVITY EXPERIMENT

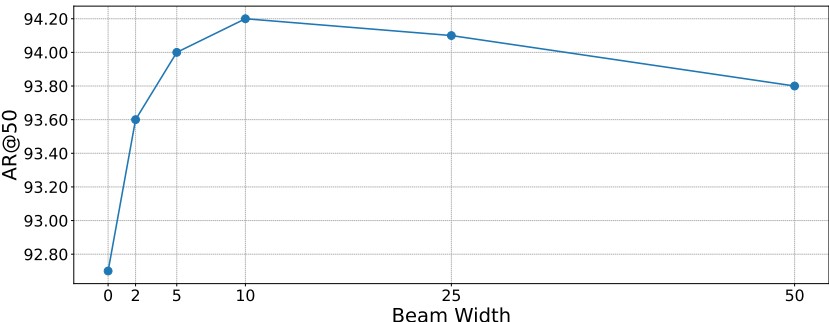

Figure 5: Change in AR@50 with varying beam width

We explored the impact of varying beam width $b$ on retrieval accuracy in terms of AR@50. The beam width directly influences the number of expanded nodes (§ 4.2). We experimented with beam widths of 0, 2, 5, 10, 25, 50 and measured the corresponding changes in AR@50.

Figure 5 illustrates the change in AR@50. We observed that AR@50 was improved by 1.7% as the beam width monotone increased from 0 to 10. This indicates that larger beam widths lead to more accurate node augmentations by performing a more exhaustive search across the expanding node space. Interestingly, when the beam size increased to 50, AR@50 decreased slightly by 0.4% compared to beam size 10. This drop may be due to LLM hallucinations in the star-based LLM refinement (SLR) module, where irrelevant edges were added to $G_l$, causing the SLR to fail in selecting the correct query-relevant nodes. This observation highlights the importance of selectively expanding only the most probable nodes within the query-relevant node expansion module.

# D QUALITATIVE ANALYSIS

**(a)** When was the **most recent Segunda Liga player of the month born**?

### SJPF Segunda Liga Player of the Month

| id | Month | Year | Nationality | Player | Team | Position |
|----|-------|------|-------------|--------|------|----------|
| 1 | May | 2009.5 2009 | Portugal | Nuno Laranjeira | Oliveirense | DF |
| 2 | September | 2009.9 2009 | Portugal | Pedro Perira | Desportivo das Aves | FW |
| 3 | October | 2009.10 2009 | Portugal | Paulo Santos | Estoril | GK |
| 4 | November | 2009.11 2009 | Portugal | Basilio Almedia | Sporting da Covilha | FW |

**Nuno Laranjeiro**
Nuno Filipe Rodrigues Laranjeiro ( born 18 January 1983 ) is a Portuguese professional footballer who plays for C.D.

**Pedro Pereira**
Pedro dos Santos Pereira ( born 28 July 1978 in Lisbon ) is a Portuguese retired footballer who played mainly as a defender.

**Paulo Santos**
Paulo Jorge da Silva dos Santos ( born 11 December 1972 ) is a Portuguese retired footballer who played as a goalkeeper.

**Portugal**
Portugal ( Portuguese : [ purtuˈɣal ] ), officially the Portuguese Republic ( Portuguese : República Portuguesa [

**Basílio Almeida**
Basilio Alexandre Neiva de Almeida (born 12 August 1971) is a Portuguese footballer who plays for Grupo Desportivo

To answer this question, we need to find the **most recent player of the month** in the table, which is the player from the latest month listed, which is **November 2009 (row 4).**
The player from row 4 is **Basílio Almeida**, who was **born on 12 August 1971**.
Therefore, the relevant rows are : [row 4]

**(b)** How many years did the series that **Zuzanna Szadkowski** appeared in for **3 episodes** run for ?

**Guiding Light**
Guiding Light ( known as The Guiding Light before 1975 ) is an American television soap opera. It is listed in Guinness World Records as the longest-running drama in television in American history, broadcast on CBS for **57 years**…

### Zuzanna Szadkowski

| Year | Title | Role | Notes |
|------|-------|------|-------|
| April | Guiding Light | Sister Angelica | 3 episodes |

**Sister Angelica**
Sister Angelica may refer to:

First, Zuzanna Szadkowski appeared in the series "Guiding Light".
Second, the linked passage titled "Guiding Light" mentions that it ran for 57 years on CBS and had a 19-year broadcast on radio.
Therefore, relevant passages are: ["Guiding Light"]

**(c)** What is the **Province where** the **Unit in the Morgan District Brigade** that **disbanded in 1782** , was founded in?

### Morgan District Brigade

| Unit | Subordinate Brigade | Created | Disbanded | Original Commander , Rank |
|------|---------------------|---------|-----------|---------------------------|
| Burke County Regiment | Salisbury , Morgan | 1777 | 1782 | Armstrong , William , Lt Col |

**2nd Rowan County Regiment**
The 2nd Rowan County Regiment was first **established in** October 22, 1775 as a local militia in Rowan County in the Province of **North-Carolina**. … On May 9, 1777, the regiment was renamed the **Burke County Regiment**, which was active until the end of the Revolutionary War in 1783

**Burke County, North Carolina**
**Burke County is a county located in the U.S. state of North Carolina.** As of the 2010 census, the population was 90,912. Its county seat is Morganton. Burke County is part of the Hickory-Lenoir-Morganton, NC Metropolitan Statistical Area.

**Montgomery County Regiment**
The Montgomery County Regiment was authorized on February 8, 1778 by the North Carolina General Assembly of 1778. It was created at the same time that Montgomery County, North Carolina was created out of the northern half of Anson County, North Carolina.

**Salisbury, North Carolina**
Salisbury is a city in the Piedmont region of North Carolina ; it has been the county seat of Rowan County since 1753 when Rowan County was much larger and its territory extended to the Mississippi River.

First, the **Unit in the Morgan District Brigade** that **disbanded in 1782** is the **Burke County Regiment**.
Second, the linked passage titled "Burke County, North Carolina" confirms that **Burke County is located in the Province of North Carolina**.
Therefore, relevant passages are: ["Burke County, North Carolina"]

**(d)** When was the first **album** of **Travie McCoy's discography** that **he guest appeared on**?

### Travie McCoy discography (Guest Appearances)

| Title | Year | Other artist (s) | Album |
|-------|------|------------------|-------|
| This Is How It Goes Down | 2008 | Pink | Funhouse |

**Pink (singer)**
Alecia Beth Moore (born September 8, 1979), known professionally as Pink (stylized as P!nk), is an American singer, songwriter, dancer, actress, record producer, spokesperson and model. She was originally a member of the girl group Choice. In 1995, LaFace Records saw potential in Pink and offered her a solo recording contract. Her R & B-influenced debut studio album Can't Take Me Home…

**Funhouse (Pink album)**
Funhouse is the fifth studio album by American singer and songwriter Pink, released by LaFace Records worldwide on **October 24, 2008**. The album debuted at number two on the Billboard 200 chart, selling 180,000 copies in its first week and reached number one on the charts in seven countries including Australia, New Zealand, Netherlands and the United Kingdom.

**Where Does This Door Go**
Where Does This Door Go is the third studio album by American singer Mayer Hawthorne. It was released on July 16, 2013, by Republic Records.

First, the question asks about the **first album** of **Travie McCoy's discography** that he **guest appeared on**.
Since the table does not specify the information about Travie McCoy, **the relevant information is not found in the given table.**
However, we can infer that the relevant information could be found in the linked passage titled **"This Is How It Goes Down"**.
It is likely that the information about the year the first album that Travie McCoy guest appeared on is mentioned in this passage.
Therefore, relevant passages are: ["This Is How It Goes Down"]

Figure 6: Qualitative analysis on four question-answer pairs. (a) A case where passage verification is successful. (b) A first case where passage verification has failed. (c) A second case where passage verification has failed. (d) A case where table aggregation is successful.

In this section, we present a qualitative analysis of FusionMaestro's Column-wise Aggregation module and Passage Verification module, with the results illustrated in Figure 6. The subfigures in Figure 6 showcase the performance and distinctive scenarios for each module: (a) highlights successful cases of the Column-wise Aggregation module, while (b), (c), and (d) demonstrate representative cases related to the Passage Verification module. For each subfigure, the query is depicted in dark blue, the data provided to submodule is shown in light blue, and the inference result from the LLM are encapsulated in a purple speech bubble with a llama icon.

Figure 6(a) shows a successful case of the column-wise aggregation module in resolving a complex query: identifying the birth date of the "most recent Segunda Liga Player of the Month." The essential part of answering this question was to recognize that the most recent player, Basilio Almeida, was honored in November 2009, as indicated in the SJPF Segunda Liga Player of the Month table. However, the initial data lacked the table segment containing the relevant row. The column-wise aggregation module reconstructed the table as shown in Figure 6(a) to include this missing information, enabling the system to restore the row with the necessary details. The LLM correctly inferred from the reconstructed table that the row corresponding to the most recent player was Row 4, based on the Year and Month columns. This lead `FusionMaestro` to accurately generate the final answer in this question, which is "12 August 1971."

Figure 6(b) shows a successful case of the passage verification module in addressing the query, "How many years did the series that Zuzanna Szadkowski appeared in for 3 episodes run for?". The module was provided with a Zuzanna Szadkowski table summarizing her appearances and a set of associated passages. The "Notes" column of the table segment confirmed that she appeared in three episodes of the series Guiding Light. The module correctly identified the one mentioning Guiding Light among the provided passages, the one which indicated that the series was broadcast on CBS for 57 years. the module correctly verified that the passage using the passage's information noting its broadcast duration, leading to an accurate answer.

Figure 6(c) shows a failure case of the Passage Verification module when answering the query, "What is the province where the unit in the Morgan District Brigade that disbanded in 1782 was founded?". The module correctly identified 'Burke County Regiment' as relevant to the query by recognizing from the 'Morgan District Brigade' table segment that the 'Disbanded' column value was 1782. However, information related to this query was present in two passages: '2nd Rowan County Regiment' and 'Burke County, North Carolina'. The LLM incorrectly verified only 'Burke County, North Carolina' as relevant, likely due to its more plausible-sounding title, while overlooking the correct answer 'North-Carolina' in the passage titled '2nd Rowan County Regiment'. Consequently, the system produced an incorrect response, 'North Carolina'. This error highlights two problems: (i) a limitation of the LLM reasoning capability and (ii) an example case of the `OTT-QA` benchmark's wrong answer annotation.

Figure 6(d) shows another failure case of the passage verification module, this time for the query, "When was the first album of Travie McCoy's discography that he guest appeared on?". Prior retrieval results correctly introduced the ground truth table titled 'Travie McCoy discography (Guest Appearances)' to the passage verification module. However, the LLM incorrectly inferred that "the table does not specify the information about Travie McCoy" as seen in the second line of its response bubble. It then relied on its parameterized knowledge to wrongly verify a passage titled 'This Is How It Goes Down as relevant'. The correct answer 'Funhouse (Pink album)' was excluded from the final retrieved document set due to the verification error.

# E    Prompts Used in Star-based LLM Refinement

Based on the prompt from `Chain-of-Table` (Wang et al.), originally used for selecting relevant rows from tables, we extended it to create column-wise aggregation and passage verification prompts, allowing for the joint consideration of table segments and linked passages.

## E.1    Prompt for Aggregation Query Classification

---

**Aggregation Query Classification**

Using `f_agg()` API, return True to detect when a natural language query involves performing aggregation operations (max, min, avg, group by). Strictly follow the format of the below examples. Please provide your explanation first, then answer the question in a short phrase starting by 'Therefore, the answer is:'

**Question**: when was the third highest paid Rangers F.C. player born?
**Explanation**: The question involves finding the birth date of the third highest paid player, which requires aggregation to find the third highest paid player. Therefore, the answer is: `f_agg([True])`

**Question**: what is the full name of the Jesus College alumni who graduated in 1960?
**Explanation**: The question involves finding the full name of the alumni who graduated in 1960, which does not require aggregation. Therefore, the answer is: `f_agg([False])`

**Question**: how tall, in feet, is the Basketball personality that was chosen as MVP most recently?
**Explanation**: The question involves finding the most recent MVP winner, which requires aggregation to identify the relevant player. Therefore, the answer is: `f_agg([True])`

**Question**: what is the highest best score series 7 of Ballando con le Stelle for the best dancer born 3 July 1969?
**Explanation**: The question involves finding the highest score in a series for a specific dancer, which requires aggregation. Therefore, the answer is: `f_agg([True])`

**Question**: which conquerors established the historical site in England that attracted 2,389,548 2009 tourists?
**Explanation**: The question involves identifying the conquerors who established a historical site, which does not require aggregation. Therefore, the answer is: `f_agg([False])`

**Question**: what is the NYPD Blue character of the actor who was born on January 29, 1962?
**Explanation**: The question involves finding the character played by an actor born on a specific date, which does not require aggregation. Therefore, the answer is: `f_agg([False])`

**Question**: '{question}'
**Explanation**:

---

918
919
920
921

## E.2 PROMPT FOR COLUMN-WISE AGGREGATION

922
923

**Column-wise Aggregation**

924
925
926
927

Using `f_row()` API to select relevant rows in the given table and linked passages that support or oppose the question. Strictly follow the format of the below example. Please provide your explanation first, then select relevant rows in a short phrase starting by: *"Therefore, the relevant rows are:"*

928
929
930
931
932
933
934
935
936
937
938
939
940
941
942
943
944
945
946
947
948
949
950
951
952
953
954
955
956
957
958
959
960

```
/* table caption : list of rangers f.c. records and
statistics
col : # | player | to | fee | date
row 1 : 1 | alan hutton | tottenham hotspur | 9,000,000 | 30
january 2008
row 2 : 2 | giovanni van bronckhorst | arsenal | 8,500,000 |
20 june 2001
row 3 : 3 | jean-alain boumsong | newcastle united |
8,000,000 | 1 january 2005
row 4 : 4 | carlos cuellar | aston villa | 7,800,000 | 12
august 2008
row 5 : 5 | barry ferguson | blackburn rovers | 7,500,000 |
29 august 2003 */
/* Passages linked to row 1:
- Alan Hutton: Alan Hutton (born 30 November 1984) is a
Scottish former professional footballer, who played as a
right back. Hutton started his career with Rangers, and won
the league title in 2005.
- Tottenham Hotspur F.C.: Tottenham Hotspur Football Club,
commonly referred to as Tottenham or Spurs, is an English
professional football club in Tottenham, London, that
competes in the Premier League. */
/* Passages linked to row 2:
- Giovanni van Bronckhorst: Giovanni Christiaan van
Bronckhorst (born 5 February 1975), also known by his
nickname Gio, is a retired Dutch footballer and currently
the manager of Guangzhou RF. */
/* Passages linked to row 3:
- Jean-Alain Boumsong: Jean-Alain Boumsong Somkong (born 14
December 1979) is a former professional football defender,
including French international.
- Newcastle United F.C.: Newcastle United Football Club is
an English professional football club based in Newcastle upon
Tyne, Tyne and Wear, that plays in the Premier League, the
top tier of English football. */
```
**Question:** 'When was the third highest paid Rangers F.C . player born ?'
**Explanation:** The third-highest paid Rangers F.C. player, Jean-Alain Boumsong (row 3). *Therefore, the relevant rows are:* `f_row([row 3])`'

961
962
963
964
965
966
967
968
969

```
/* '{table}' */

/* '{linked_passages}' */
```

970
971

**Question:** '{question}'
**Explanation:**

### E.3 PROMPT FOR PASSAGE VERIFICATION

**Passage Verification**

Using `f_passage()` API to return a list of passage titles that are relevant to the question, even if they are only partially related. Strictly follow the format of the below example. Please provide your explanation first, then return a list of passages in a short phrase starting by: *"Therefore, relevant passages are:"*

```
/* table caption : List of politicians, lawyers, and civil
servants educated at Jesus College, Oxford
col : Name | M | G | Degree | Notes
row 1 : Lalith Athulathmudali | 1955 | 1960 | BA
Jurisprudence (2nd, 1958), BCL (2nd, 1960) | President of
the Oxford Union (1958); a Sri Lankan politician; killed by
the Tamil Tigers in 1993 */
/* List of linked passages: ["Law degree", "Oxford Union",
"Lalith Athulathmudali"]
Title: Lalith Athulathmudali. Content: Lalith William
Samarasekera Athulathmudali, PC (Sinhala; 26 November 1936
- 23 April 1993), known as Lalith Athulathmudali, was a Sri
Lankan statesman. He was a prominent member of the United
National Party, who served as Minister of Trade and Shipping;
Minister of National Security and Deputy Minister of Defence;
Minister of Agriculture, Food and Cooperatives, and finally
Minister of Education.
Title: Law degree. Content: A law degree is an academic
degree conferred for studies in law. Such degrees are
generally preparation for legal careers; but while their
curricula may be reviewed by legal authority, they do not
themselves confer a license. A legal license is granted
(typically by examination) and exercised locally; while the
law degree can have local, international, and world-wide
aspects.
Title: Oxford Union. Content: The Oxford Union Society,
commonly referred to simply as the Oxford Union, is a
debating society in the city of Oxford, England, whose
membership is drawn primarily from the University of Oxford.
Founded in 1823, it is one of Britain's oldest university
unions and one of the world's most prestigious private
students' societies. The Oxford Union exists independently
from the university and is separate from the Oxford
University Student Union. */
```

**Question:** What is the full name of the Jesus College alumni who graduated in 1960?
**Explanation:** First, Lalith Athulathmudali graduated in 1960. Second, the linked passage titled "Lalith Athulathmudali" confirms his full name. *Therefore, relevant passages are:* `f_passage(["Lalith Athulathmudali"])`

```
/* '{table}' */

/* '{linked_passages}' */
```

**Question:** '{question}'
**Explanation:**