# OpenReview forum: "FusionMaestro: Harmonizing Early Fusion, Late Fusion, and LLM Reasoning for Multi-Granular Table-Text Retrieval"
_ICLR.cc/2025/Conference — Submitted to ICLR 2025_

### Official Review · Reviewer_LAen · 2024-10-17

**Soundness:** 3
**Presentation:** 3
**Contribution:** 3
**Rating:** 6
**Confidence:** 3

**Summary:**

This paper constructs a novel framework, FusionMaestro, for the table-text task. The method involves a well-designed pipeline that combines the advantages of early fusion and late fusion to generate more accurate table-text retrieval results. Experimental results confirm that FusionMaestro significantly outperforms existing baselines in both table-text retrieval accuracy and end-to-end response accuracy. Besides, the experimental results also proves the contribution of each process in the pipeline and validates the effectiveness of the proposed edge-based information granularity in improving results.

**Strengths:**

* Although the method and pipeline proposed in the paper are very complex and involve many details, the authors have meticulously written the paper, ensuring logical consistency and smooth readability. The writing level of this article is relatively high.
* Experimental results demonstrate that the proposed method has significant advantages compared to existing early fusion and late fusion methods.
* The experimental analysis is comprehensive and solid, examining the effectiveness of each component proposed.

**Weaknesses:**

The designed framework is very complex, with intricate steps and a vast amount of detailed information. Although the authors have provided open-source code, this poses a certain level of challenge to understanding the method proposed in this paper.

**Questions:**

I noticed that the proposed pipeline involves the utilizing LLM to refine the graph information. It can be seen that this step essentially leverages the capabilities of LLM to enhance the overall effectiveness of the pipeline. Is this fair compared to the existing baselines?

---

> ### Author Response · Authors · 2024-11-21
> **Response to Reviewer LAen**
>
> **Response to W1:** Thank you for your valuable feedback. We would like to introduce a brief review of our approach to guide you along the overall process, followed by an explanation of how we have addressed your comments in our paper.
>
> Our approach is composed of three stages.
> (i) In the **Edge-based Bipartite Subgraph Retrieval** stage, we use edges as the basic retrieval unit within a bipartite data graph generated by early fusion. This choice avoids including query-irrelevant nodes, such as fusion blocks, and provides richer context than single nodes. The retrieved edges are integrated into a single bipartite subgraph.
> (ii) The **Query-relevant Node Expansion** stage strengthens the subgraph by identifying promising seed nodes within the retrieved graph for additional hops. Expanded queries are then used to locate candidate nodes, and the most relevant candidates are added to the graph.
> (iii) Finally, in the **Star-based LLM Refinement** stage, the graph undergoes further improvement using the reasoning capabilities of LLMs. This involves aggregation for query-related tables and verification of query-irrelevant passages. The refined graph is decomposed into a ranked sequence of edges, completing the framework.
>
> To improve the readability of the paper, we had two graduate students proofread the manuscript and provide us with detailed feedback. Specifically, we  have revised Sections 3 and 4 with the following revisions, and have  highlighted them in blue in the revised manuscript:
> - Section 3: we added explanations for newly defined notations, including how terms such as $G_{init}$ correspond to elements in the initial problem formulation.
> - Section 4 (Overview): we rephrased the descriptions of framework components to ensure each module serves as the subject of its respective explanation. Additionally, removed notations (e.g., $G_q$, $E_q$) that did not contribute to understanding.
> - Section 4.3 (Final Paragraph): we provided a detailed explanation of top-K edge selection process, clarifying the transition from the retrieved graph to the ranked output.
>
> **Response to Q1:** Simply employing an LLM does not guarantee improved performance in table-text retrieval. In the ablation study in Section 5.4, we conducted experiments using two refinement units with an LLM, comparing the performance of a full graph prompt to individual prompts for each star graph. The impact of retrieval unit granularity on accuracy results in an nDCG@50 score that is 12.4% higher than the full graph setting.
>
> FusionMaestro outperforms COS by an average of 17.6% in AR@k across all k values even when Star-based LLM Refinement (SLR) is excluded. Furthermore, it achieves a 38.4% improvement over COS in nDCG@50. This highlights the contributions of our non-LLM-related components, namely (1) the combined usage of early and late fusion, and (2) the granularity determination strategy tailored for each retrieval stage. The role of the LLM is to further enhance results by refining already strong retrieval outcomes rather than compensating for its fundamental weaknesses. We have included detailed ablation results in Section 5.3 of our paper to illustrate these points.
>
> | **Model**                   | **AR@2**         | **AR@5**         | **AR@10**        | **AR@20**        | **AR@50**        | **nDCG@50**       |
> |--------------------------|--------------|--------------|--------------|--------------|--------------|---------------|
> | **COS**                     | 44.4         | 61.6         | 70.8         | 79.5         | 87.8         | 33.6          |
> | **FusionMaestro w/o SLR**    | 60.0 (+35.1%)| 75.2 (+22.1%)| 84.7 (+19.6%)| 90.1 (+13.3%)| 94.6 (+7.7%) | 46.5 (+38.4%) |

---

> > ### Comment · Reviewer_LAen · 2024-12-01
> >
> > Thank you for your detailed response for my comments. I decide to maintain my score.

---

> > > ### Author Response · Authors · 2024-12-01
> > > **Sincere Gratitude from Authors**
> > >
> > > Thank you for considering our responses in detail. Your constructive comments have been instrumental in refining and enhancing our work.

---

### Official Review · Reviewer_fKL9 · 2024-10-31

**Soundness:** 2
**Presentation:** 2
**Contribution:** 2
**Rating:** 6
**Confidence:** 3

**Summary:**

In table-text open-domain question answering, a retriever system must retrieve relevant evidence from both tables and text to accurately answer questions. The core challenge in table-text retrieval lies in integrating these two information formats, as textual data is unstructured, while tabular data is highly structured. Existing methods are categorized into two main approaches based on the timing of table and text integration: early fusion and late fusion. Early fusion aligns table rows with associated passages in advance using entity linking, which can overlook query-dependent relationships and include irrelevant context. In contrast, late-fusion dynamically aligns relevant table rows and passages after the query is given, which, however, risks missing relevant contexts.
This paper introduces a new table-text retriever framework, FusionMaestro, which combines both early- and late-fusion strategies. FusionMaestro formalizes the retrieval task as a bipartite graph problem, representing table segments and passages as two disjoint node sets and the links between them as edges. The goal is to identify query-relevant edges within this graph. The pipeline consists of four main stages:
1. Early Fusion: Links are established between text and table data via entity linking to generate an initial graph.
2. Edge-based Bipartite Subgraph Retrieval: Each edge in the graph is embedded based on its connected table and text, while the query is also embedded, and relevant edges are then retrieved to form a subgraph.
3. Query-Relevant Node Expansion: Nodes are embedded, and seed nodes are identified based on their relevance to the query embedding. These seed nodes are then expanded along their edges within the graph.
4. Star-Based LLM Refinement: An LLM identifies the most query-relevant table segments or passages in the graph, filtering out irrelevant nodes.
The refined edge-scored graph ultimately provides a ranked sequence for retrieval results. Experiments on the OTT-QA benchmark demonstrate that this framework significantly improves retrieval performance.

**Strengths:**

1. The paper first formalizes the retrieval task as a bipartite graph problem, which is an interesting and reasonable approach that clarifies the task.
2. Building on bipartite graph modeling, FusionMaestro effectively combines early- and late-fusion methods. It begins with early-fusion to create an initial graph and then uses edges as the basic retrieval unit within this bipartite structure. This is a strong approach that leverages the advantages of both fusion methods: it selects query-relevant tables and text while integrating semantic information from both text and corresponding tables through early fusion, ultimately improving retrieval accuracy.
3. The proposed method employs prompts to instruct the LLM to aggregate table information and identify documents relevant to the query. By incorporating an LLM, this approach effectively enhances retrieval for queries that demand complex reasoning, such as multi-hop retrieval or column-wise aggregation.

**Weaknesses:**

1. The experiments involve different types of retrievers, which may lead to an unfair comparison. The proposed method uses ColBERTv2, a late-interaction retriever, which is more resource-intensive than single-embedding retrievers. However, none of the baselines use late-interaction retrievers. Given that the primary contribution of this paper is the retrieval framework itself, experiments should evaluate a single-embedding retriever to ensure consistency with the baselines and to more effectively validate the framework's effectiveness.
2. The paper does not provide many details of the proposed method. (a) **Edge weight calculation**: The final refined edge-scored graph $G_q$ generated by LLM lacks an explanation of how edge weights are calculated. Since the edge weights of $G_q$ are critical for the final output, this is a crucial part of the method. (b) **Role of edge weights**: Each stage in this method generates a graph with edge weights, yet the paper does not explain the purpose of these intermediate edge weights. It appears that edge weights before the “LLM Refinement” step are unused. (c) **Output results**: The authors state that “the refined edge-scored graph $G_q$ is decomposed into a ranked sequence of edges $\epsilon_q$ and returned as the result.” However, the paper does not explain how this decomposition is performed, leaving the reader uncertain about how the final output is generated.
3. Although the experimental results indicate that the proposed method performs well, it is significantly more time- and resource-intensive compared to the baselines. This approach requires the use of ColBERT and Llama-3.1-8B-Instruct, along with multiple rounds of LLM inference, which is highly demanding in terms of both computational resources and time.

**Questions:**

The questions are listed in the 'Weaknesses' section. Another question is whether you recorded the inference times of the methods in the experiments and if you could provide retrieval time results for the proposed method compared to the baselines.

---

> ### Author Response · Authors · 2024-11-21
> **Response to Reviewer fKL9 (1/2)**
>
> **Response to W1:** Thank you for your valuable feedback. While we acknowledge that retrievers and rerankers can enhance accuracy, our primary goal is to demonstrate that the three components of FusionMaestro are agnostic to specific retrievers and rerankers. To support this, we conducted additional experiments using single-embedding retrievers and the reranker from COS. These models were fine-tuned for edge-level retrieval using a contrastive learning approach to ensure a fair consistent comparison.
>
> The experimental results confirm that FusionMaestro remains robust and outperforms COS, even when restricted to using the COS retrievers and reranker, highlighting its retriever- and reranker-agnostic capabilities. Specifically, FusionMaestro outperformed COS by an average of 7.0% in AR@k across all k values. This improvement is also reflected in nDCG@50, where FusionMaestro achieved a 9.2% gain over COS. Even without the Star-based LLM Refinement (SLR) module, FusionMaestro showed a 4.2% average improvement in AR@k and a 7.1% improvement in nDCG@50 compared to COS.
>
> | **Model**                                     | **AR@2**         | **AR@5**         | **AR@10**        | **AR@20**        | **AR@50**        | **nDCG@50**       |
> |-------------------------------------------|--------------|--------------|--------------|--------------|--------------|---------------|
> | **COS**                                    | 44.4         | 61.6         | 70.8         | 79.5         | 87.8         | 33.6          |
> | **FusionMaestro with COS’ Dense Retrievers & Reranker** | 51.4 (+15.8%)| 68.2 (+10.7%)| 77.1 (+8.9%) | 83.0 (+4.4%) | 88.4 (+0.7%) | 36.7 (+9.2%)  |
> | **w/o SLR**                                   | 45.6 (+2.7%) | 65.5 (+6.3%) | 76.0 (+7.3%) | 83.1 (+4.5%) | 88.3 (+0.6%) | 36.0 (+7.1%)  |
>
> **Response to W2:** We have carefully addressed your concerns regarding (a) the method of LLM’s edge weight calculation, (b) the role of edge weights, and (c) the decomposition of the final output. Each point is detailed in the bullets below.
> - (a) The LLM does not calculate or assign additional edge scores; rather, it performs a binary verification to determine whether an edge is relevant to the query. This verification process identifies and removes query-irrelevant edges while preserving the existing edge scores from earlier stages. To make this clear, we have revised Section 4.3 to explicitly state that the LLM conducts binary verification and does not recalculate edge scores.
> - (b) You are correct in observing that intermediate edge scores appear to be unused before the LLM refinement step. These scores are used only in the last step of FusionMaestro for linearizing the final retrieved graph, where we have to determine the order in which edges are provided to the reader LLM.
> In Section 3.2, we have clarified the role of edge scores when defining our graph data structure. In Section 4.1, we have added a clarification when assigning the initial edge scores to the candidate bipartite subgraph $G_c$​ to prevent confusion. In Section 4.3, we have explained that the assigned edge scores are not utilized during the intermediate steps but are instead used in the graph decomposition described.
> - (c) The term "graph decomposition process" has been updated to Top-K Edge Selection, which involves sorting all edges by their scores in descending order and selecting the top-ranked edges as required.

---

> > ### Author Response · Authors · 2024-11-21
> > **Response to Reviewer fKL9 (2/2)**
> >
> > **Response to W3 & Q1:** We have conducted additional experiments to measure the inference times and analyzed the efficiency of our proposed method compared to the baselines.
> >
> > To review FusionMaestro, it consists of three novel key components: edge-based bipartite subgraph retrieval, query-relevant node expansion (QNE), and star-based LLM refinement (SLR). Among these, the SLR module is the primary efficiency bottleneck. We want to point out that our approach significantly outperforms state-of-the-art methods such as CORE and COS in retrieval accuracy, even when the SLR module is excluded, while maintaining competitive speed. Incorporating the SLR module with an LLM further enhances performance, though it introduces a trade-off between computational overhead and accuracy gains. As advancements in LLM inference efficiency continue, we anticipate that this trade-off will shift favorably, making the enhanced retrieval accuracy offered by the SLR module more accessible.
> >
> > We measured the retrieval times for two versions of FusionMaestro with and without the SLR module under identical conditions. All experiments were conducted on 4 RTX A6000 GPUs, with LLM inference managed via the SGLang inference engine.
> >
> > | **Model**               | **Retrieval time per query (s)** | **AR@2**        |
> > |--------------------------|-----------------------------------|-----------------|
> > | **CORE**                | 6.02                             | 35.3           |
> > | **COS**                 | 4.89                             | 44.4           |
> > | **FusionMaestro**       | 10.74 (+119.6%)                  | 63.3 (+42.6%)  |
> > | **FusionMaestro w/o SLR** | 5.15 (+5.3%)                    | 60.0 (+35.1%)  |
> >
> > The results demonstrate that without the SLR module, our model achieves a retrieval speed highly competitive with COS,  with only a 5.3% increase in runtime (outperforming CORE by 16.9%) while delivering a substantial 35.1% improvement in AR@2. A breakdown of the retrieval process excluding SLR shows that edge-based bipartite subgraph retrieval and QNE modules contribute 49% and 51%, respectively. When the SLR module is included, it introduces approximately 5.59 seconds of overhead, resulting in a 5.5% improvement in AR@2 performance. However, this overhead can be substantially reduced through advancements in hardware, such as H100, and software optimization techniques.

---

> > > ### Comment · Reviewer_fKL9 · 2024-11-25
> > >
> > > Thank you for your response. My concerns have been addressed. Therefore, I have increased the overall rating from 5 to 6.

---

> > > > ### Author Response · Authors · 2024-11-25
> > > > **Sincere Gratitude from Authors**
> > > >
> > > > Thank you for taking the time to review our responses and for revising your evaluation. We are delighted to hear that we have addressed your concerns, and we deeply value your insightful feedback, which has greatly contributed to improving our work.

---

### Official Review · Reviewer_WRaF · 2024-11-01

**Soundness:** 3
**Presentation:** 3
**Contribution:** 3
**Rating:** 6
**Confidence:** 4

**Summary:**

This paper proposes a novel graph-based approach for table retrieval, named FusionMaestro. The authors first examine the limitations of previous work, noting that early fusion methods, which are query-independent, risk including irrelevant passages, while late fusion methods may only retrieve partial relevant information. Both fusion approaches also primarily rely on semantic matching, making it challenging to answer questions that require complex reasoning.

FusionMaestro aims to address these limitations by combining the strengths of early and late fusion methods. In this approach, early fusion is used to construct an initial graph that links tables to passages, with edges as the retrieval unit. A two-stage ranking mechanism then selects the most relevant edges to the query. During late fusion, the most relevant nodes are identified as seed nodes, and neighboring nodes are integrated into the graph. Finally, an LLM is used for graph refinement, performing column-wise aggregatiol and verifying passages to filter out irrelevant edges. FusionMaestro’s output is a set of ranked edges from the final retrieved graph. Experimental results demonstrate that this method significantly outperforms baselines on two benchmark datasets.

**Strengths:**

- S1: The paper is well-written, and the approach is clearly motivated, effectively combining the advantages of both early and late fusion methods.
- S2: Reformulating table retrieval as a graph retrieval task is a unique perspective. Using edges as the retrieval unit adds an interesting new approach to the task.
- S3: The authors conduct comprehensive experiments on two benchmark datasets, showing that the proposed method effectively retrieves relevant table information, thereby enhancing table QA performance.

**Weaknesses:**

- W1: The edge retriever and reranker are critical components, yet the authors do not evaluate their method against different ranking models (e.g., sparse ranking or state-of-the-art dense retrievers).
- W2: The authors note that removing the star-based LLM refinement increased AR@50 but attribute this to LLM hallucination without providing qualitative analysis. Including a representative example or conducting quantitative analysis (e.g., with off-the-shelf hallucination detection models) would strengthen the findings.
- W3: Table retrieval has been extensively explored in information retrieval, but some related work on graph-based methods for table retrieval [1,2] is missing from the references.


[1] Retrieving Complex Tables with Multi-Granular Graph Representation Learning, SIGIR 2021

[2] MGNETS: Multi-Graph Neural Networks for Table Search, CIKM 2021

**Questions:**

See Weaknesses.

---

> ### Author Response · Authors · 2024-11-21
> **Response to Reviewer WRaF**
>
> **Response to W1:** Thank you for your insightful question. While improvements in retrievers and rerankers can enhance accuracy, we aim to demonstrate that the three components of FusionMaestro are agnostic to specific retrievers and rerankers. To support this, we conducted additional experiments using different retrievers and reranker, specifically dense retrievers and the reranker from COS. These models were fine-tuned for edge-level retrieval using a contrastive learning approach to ensure a fair and consistent comparison.
>
> The experimental results show that FusionMaestro remains robust and outperforms COS, even when restricted to using the COS retrievers and reranker, highlighting its retriever- and reranker-agnostic capabilities.  Specifically, FusionMaestro outperformed COS by an average of 7.0% in AR@k across all k values. This improvement is also reflected in nDCG@50, where FusionMaestro achieved a 9.2% gain over COS. Even without the Star-based LLM Refinement (SLR) module, FusionMaestro showed a 4.2% average improvement in AR@k and a 7.1% improvement in nDCG@50 compared to COS.
>
> | **Model**                                     | **AR@2**         | **AR@5**         | **AR@10**        | **AR@20**        | **AR@50**        | **nDCG@50**       |
> |-------------------------------------------|--------------|--------------|--------------|--------------|--------------|---------------|
> | **COS**                                    | 44.4         | 61.6         | 70.8         | 79.5         | 87.8         | 33.6          |
> | **FusionMaestro with COS’ Dense Retrievers & Reranker** | 51.4 (+15.8%)| 68.2 (+10.7%)| 77.1 (+8.9%) | 83.0 (+4.4%) | 88.4 (+0.7%) | 36.7 (+9.2%)  |
> | **w/o SLR**                                   | 45.6 (+2.7%) | 65.5 (+6.3%) | 76.0 (+7.3%) | 83.1 (+4.5%) | 88.3 (+0.6%) | 36.0 (+7.1%)  |
>
> **Response to W2:** We have conducted both quantitative and qualitative analysis to investigate the reasons behind the decrease in retrieval accuracy (AR@50) when employing the star-based LLM refinement (SLR) module. Specifically, we observed 12 instances where the LLM failed to select the correct passage despite being provided with the ground truth passage. Upon manual examination of these cases, we found that 10 of them were attributable to hallucinations, where the LLM performed incorrect reasoning. In the remaining 2 cases, the LLM correctly identified the relevant document, but this document was not included in the provided annotations.
>
> In addition to this quantitative analysis, we selected representative examples from both categories to perform a detailed qualitative analysis. To provide a more comprehensive analysis, we also examined cases where the column-wise aggregation and passage verification steps led to successful reasoning. We have included these representative examples in Appendix D: Qualitative Analysis Section of the revised manuscript for your reference.
>
> **Response to W3:** Thank you for introducing the references related to graph-based methods for table retrieval. While we acknowledge the contributions of the proposed references, they address a problem that is orthogonal to our focus.  The proposed references [1, 2] introduce approaches that leverage graph neural networks to encode the structural information of tables effectively. On the other hand, FusionMaestro centers on retrieving both tables and text upon the semantic relationships between them. Integrating improved encoding of structure data into our system would be interesting future work.
>
> To clarify this distinction of these graph-based methods, we have revised the Related Work Section (§2.2: Table-Text Retrieval) in our manuscript. We explicitly discussed the differences between our work and the suggested references to ensure readers understand how these methods relate to, yet diverge from, our problem scope.

---

> > ### Comment · Reviewer_WRaF · 2024-11-23
> >
> > Thank you for your response; my concerns are addressed.

---

> > > ### Author Response · Authors · 2024-11-25
> > > **Sincere Gratitude from Authors**
> > >
> > > We sincerely appreciate your positive feedback and are grateful for the time and effort you devoted to reviewing our work. Your thoughtful suggestions have been invaluable in enhancing the quality of our paper.

---

### Official Review · Reviewer_iPHZ · 2024-11-02

**Soundness:** 2
**Presentation:** 3
**Contribution:** 2
**Rating:** 6
**Confidence:** 4

**Summary:**

This paper proposes FusionMaestro, a novel method to table-text retrieval in open-domain question answering. FusionMaestro leverages both early fusion and late fusion techniques, combined with large language model(LLM) reasoning, to address column-wise aggregation and multi-hop reasoning. Specifically, FusionMaestro utilizes a multi-granular retrieval pipeline, integrating edge-based bipartite subgraph retrieval, query-relevant node expansion, and star-based LLM refinement, resulting in significant performance improvements over state-of-the-art models. Additionally, the authors validate the effectiveness of their approach on the OTT-QA and MMQA datasets and conduct ablation studies to demonstrate the contribution of each component.

**Strengths:**

S1. FusionMaestro combines early and late fusion techniques while utilizing the reasoning capabilities of large language models to refine retrieval results,  effectively addressing the limitations of previous approaches, particularly in handling complex reasoning tasks, and improving overall performance.
S2. FusionMaestro constructs bipartite subgraph using edges as the retrieval unit and further expands these with query-relevant nodes, providing richer contextual information and enhancing both the accuracy and comprehensiveness of retrieval.
S3. The experiments in this paper cover multiple datasets and baselines, along with ablation studies, effectively validating the effectiveness of FusionMaestro.

**Weaknesses:**

W1. Although the proposed method achieves impressive performance, the combination of multiple components, finer-grained retrieval, and the use of large language models undoubtedly introduce significant computational overhead and complexity. However, the paper lacks an analysis of efficiency to address these concerns.
W2. Although the paper includes an ablation study, the evaluation section only provides AR@k and nDCG@50 scores. Reporting end-to-end performance would be more intuitive, as it would help illustrate the performance gains achieved in relation to the increased computational costs.
W3. Among the baselines compared with FusionMaestro, were any methods that utilized LLMs or LLM-like architectures? If so, for fairness, should the same LLM have been used across comparisons?

**Questions:**

see weakness

---

> ### Author Response · Authors · 2024-11-21
> **Response to Reviewer iPHZ**
>
> **Response to W1:** Thanks for your insightful questions. Our method integrates three novel concepts: edge-based bipartite subgraph retrieval, query-relevant node expansion (QNE), and star-based LLM refinement (SLR). Among these, SLR is the primary efficiency bottleneck. Importantly, our approach significantly outperforms the state-of-the-art methods, CORE and COS in retrieval accuracy, even without SLR, while maintaining competitive speed. Incorporating SLR with an LLM further boosts performance, though it introduces a trade-off between computational overhead and performance gains. As advancements in LLM inference efficiency continue to accelerate, we anticipate that this trade-off will increasingly favor the adoption of SLR, making the enhanced retrieval accuracy more accessible.
>
> To address concerns about efficiency, we provide a detailed analysis of retrieval time with and without the SLR module, compared against CORE and COS. To ensure a fair comparison, all experiments were conducted under identical conditions using 4 RTX A6000 GPUs, with LLM inference managed via the SGLang inference engine.
>
> | **Model**               | **Retrieval time per query (s)** | **AR@2**        |
> |--------------------------|-----------------------------------|-----------------|
> | **CORE**                | 6.02                             | 35.3           |
> | **COS**                 | 4.89                             | 44.4           |
> | **FusionMaestro**       | 10.74 (+119.6%)                  | 63.3 (+42.6%)  |
> | **FusionMaestro w/o SLR** | 5.15 (+5.3%)                    | 60.0 (+35.1%)  |
>
> The results demonstrate that without the SLR module, our model achieves a retrieval speed highly competitive with COS,  with only a 5.3% increase in runtime (outperforming CORE by 16.9%) while delivering a substantial 35.1% improvement in AR@2. A breakdown of the retrieval process excluding SLR shows that edge-based bipartite subgraph retrieval and QNE modules contribute 49% and 51%, respectively. When the SLR module is included, it introduces approximately 5.59 seconds of overhead, resulting in a 5.5% improvement in AR@2 performance. However, this overhead can be substantially reduced through advancements in hardware, such as H100, and software optimization techniques.
>
> **Response to W2:** We have conducted additional experiments as part of the ablation study, incorporating Exact Match (EM) and F1 scores as end-to-end evaluation metrics. Specifically, we evaluated two variants of FusionMaestro by excluding each of the Query-relevant Node Expansion (QNE) and Star-based LLM Refinement (SLR) modules, while providing 50 edges as input to the reader.
>
>
> | **Model**                | **EM**       | **F1**       | **AR@50**          | **nDCG@50**          |
> |---------------------------|--------------|--------------|--------------------|----------------------|
> | **FusionMaestro**        | 59.3         | 65.8         | 94.2               | 47.0                 |
> | **w/o QNE**              | 56.9 (-4.2%) | 63.2 (-4.1%) | 92.7 (-1.6%)       | 45.1 (-4.2%)         |
> | **w/o SLR**              | 59.0 (-0.5%) | 65.7 (-0.2%) | 94.6 (+0.4%)       | 46.5 (-1.1%)         |
>
> Our additional experiments demonstrate that the retrieval accuracy trends observed in the AR@k and nDCG@50 scores are consistent with those in the end-to-end EM and F1 scores. For example, excluding the QNE module resulted in a 4.2% reduction in EM and a 4.1% reduction in F1 scores, while excluding the SLR module led to reductions of 0.5% and 0.2% in EM and F1 scores, respectively. These trends align closely with the reductions observed in nDCG@50 (4.2% and 1.1%, respectively). Notably, while the AR@50 score increased by 0.4% when the SLR module was excluded, this divergence suggests that nDCG@50 better reflects the degree to which the retrieved information is necessary for answering the query accurately. We have incorporated these results into the revised manuscript, specifically expanding Table 4 in §5.3 to include columns for EM and F1 scores.
>
> **Response to W3:** To the best of our knowledge, **FusionMaestro** is the first method to apply LLMs to the *table-text retrieval task*. None of the baseline methods we compared against utilized LLMs or LLM-like architectures.

---

> > ### Comment · Reviewer_iPHZ · 2024-11-26
> >
> > Thanks for your response. The response has addressed my concern.

---

> > > ### Author Response · Authors · 2024-11-26
> > > **Sincere Gratitude from Authors**
> > >
> > > Thank you for your response. We are pleased to hear that our reply has addressed your concerns. Your insightful suggestions have been instrumental in enhancing the quality of our paper.
> > >
> > > Please let us know if there is anything further we need to address to help you consider updating your score.

---

> > > > ### Comment · Reviewer_iPHZ · 2024-11-26
> > > >
> > > > I have increased the score.

---

> > > > > ### Author Response · Authors · 2024-11-26
> > > > > **Sincere Gratitude from Authors**
> > > > >
> > > > > Thank you so much for your positive response and for updating the score!

---

### Meta-Review · Area_Chair_EQ3h · 2024-12-20

**Metareview:**

This paper introduces a new method for hybrid retrieval of tables and text, formulated as a node retrieval task on a graph consisting of tables and text passages. The approach leverages graph-based retrieval with both early and late fusion and employs language model reasoning capabilities to refine initial retrieval results. Experiments on the OTT-QA and MMQA datasets show significant improvements in both retrieval accuracy and final QA performance.

Strengths
- Introduction of the new method for table-text retrieval with early and late fusion (iPHZ, WRaF, fKL9)
- Reformulating table retrieval as a graph retrieval is valuable (WRaF, fKL9).
- Experiments are comprehensive and demonstrate impressive results and analysis (iPHZ, WRaF, LAen)
- The paper is well written (​​LAen).

Weaknesses
- The use of LM for refinement leads to significant increase in computational cost and methodological complexity (iPHZ, fKL9, LAen).
    - The authors' responses acknowledge that the full model significantly increases latency (+119.6%) for a +42.6% improvement. However, without SLR, latency remains nearly unchanged (+5.3%) while still achieving substantial improvements (+35.1%).
- Lack of ablations on the choice of retriever and reranker, especially because (1) the proposed model makes use of more complex and computationally expensive retrieval methods like ColBERT, and (2) the proposed model makes use of LLM that is more expensive, and it is unclear whether the gains are simply coming from the use of such expensive models or truly coming from the proposed changes in model architectures (iPHZ, fKL9, WRaF)
    - The author responses provide additional experiments that show that, with retrieval/reranker same as COS (the baseline), there is still improvements, e.g., COS achieves 33.6, FusionMaestro with COS’ retrieval and reranker achieves 36.0, and the original FusionMaestro achieves 47.0, and nDCG@50 on OTT-QA.
    - However, this ablation study does not report on end-to-end QA accuracy, arguably the more important metric. Notably, COS achieves 61.5 F1, while the original FusionMaestro achieves 64.3 F1 on OTT-QA. This suggests that FusionMaestro, when using the same retrieval/reranking as COS, likely achieves an F1 score between 61.5 and 64.3.

Authors are encouraged to address these two points in the next iteration of the paper, given that they are common concerns raised by reviewers, and to include comprehensive and controlled ablations on both retrieval and end QA accuracy.

**Additional Comments On Reviewer Discussion:**

Weaknesses resolved during rebuttal
- Lack of ablations/discussion (iPHZ, fKL9, WRaF, LAen) -> additional results provided as noted in the previous section
- Lack of end-to-end QA accuracy metrics (iPHZ)
- Lack of discussion on related work (iPHZ)
- Lack of details (fKL9)

---

### Decision · Program_Chairs · 2025-01-22

Reject